**Timescales of Secondary Organic Aerosols to Reach Equilibrium at**
**Various Temperatures and Relative Humidities**
**Ying Li[1], and Manabu Shiraiwa[1,*]**
[1] Department of Chemistry, University of California, Irvine, California, USA.
[*]Correspondence to: Manabu Shiraiwa (m.shiraiwa@uci.edu)
**Abstract:**
Secondary organic aerosols (SOA) account for a substantial fraction of air particulate
matter and SOA formation is often modeled assuming rapid establishment of
gas-particle equilibrium. Here, we estimate the characteristic timescale for SOA to
achieve gas−particle equilibrium under a wide range of temperatures and relative
humidities using a state-of-the-art kinetic flux model. Equilibration timescales were
calculated by varying particle phase state, size, mass loadings, and volatility of
organic compounds in open and closed systems. Model simulations suggest that the
equilibration timescale for semi-volatile compounds is on the order of seconds or
minutes for most conditions in the planetary boundary layer, but it can be longer than
one hour if particles adopt glassy or amorphous solid states with high glass transition
temperature at low relative humidity. In the free troposphere with lower temperatures
it can be longer than hours or days even at moderate or relatively high relative
humidity due to kinetic limitations of bulk diffusion in highly viscous particles. The
timescale of partitioning of low-volatile compounds into highly viscous particles is
shorter compared to semi-volatile compounds in the closed system, as it is largely
determined by condensation sink due to very slow re-evaporation with relatively
quick establishment of local equilibrium between the gas phase and the near-surface
bulk. The dependence of equilibration timescales on both volatility and bulk
diffusivity provides critical insights into thermodynamic or kinetic treatments of SOA
partitioning for accurate predictions of gas- and particle-phase concentrations of
semi-volatile compounds in regional and global chemical transport models.

## 1. Introduction

Secondary organic aerosols (SOA) play a central role in climate, air quality and public health. Accurate descriptions of formation and evolution of SOA remain a grand challenge in climate and air quality models (Kanakidou et al., 2005; Shrivastava et al., 2017a). Current chemical transport models usually employ instantaneous equilibrium partitioning of semi-volatile oxidation products into the particle phase (Pankow, 1994), assuming that SOA partitioning is rapid compared to the timescales of other major atmospheric processes associated with SOA formation. The timescale of SOA to reach equilibrium with their surrounding condensable vapors needs to be evaluated under different ambient conditions to validate this assumption.

SOA particles can adopt liquid (dynamic viscosity $\eta < 10^2$ Pa s), semi-solid ($10^2 \leq \eta \leq 10^{12}$ Pa s), or glassy or amorphous solid states ($\eta > 10^{12}$ Pa s), depending on chemical composition, temperature ($T$) and relative humidity (RH) (Virtanen et al., 2010; Koop et al., 2011; Zhang et al., 2015; Reid et al., 2018). The occurrence of glassy or amorphous solid states may lead to kinetic limitations and prolonged equilibration timescale in SOA partitioning (Shiraiwa and Seinfeld, 2012; Booth et al., 2014; Zaveri et al., 2014; Mai et al., 2015), affecting evolution of particle size distribution upon SOA growth (Maria et al., 2004; Shiraiwa et al., 2013a; Zaveri et al., 2018). A number of experimental studies have indeed observed kinetic limitations of the bulk diffusion of organic molecules (Vaden et al., 2011; Perraud et al., 2012; Ye et al., 2016a; Zhang et al., 2018), while chamber experiments probing the

intraparticle mixing did not find kinetic limitations at moderate and high RH and
room temperature (Ye et al., 2016b; Gorkowski et al., 2017; Ye et al., 2018).
Recently, global simulations predicted that SOA particles are expected to be
mostly in a glassy solid phase state in the middle and upper troposphere and also in
dry lands in the boundary layer (Shiraiwa et al., 2017), which can lead to prolonged
characteristic bulk diffusion timescales of organic molecules within SOA particles
(Shiraiwa et al., 2011; Maclean et al., 2017). Slow bulk diffusion associated with a
glassy phase state can prevent atmospheric oxidants to react with organic compounds
such as polycyclic aromatic hydrocarbons (Shrivastava et al., 2017b; Mu et al., 2018),
contributing to long-range transport of organic compounds. Recent ambient
observations have shown that the condensation of highly oxygenated molecules
(HOMs), which play an important role in new particle formation, would be governed
by kinetic partitioning in the free troposphere (Bianchi et al., 2016). Diffusivity
measurements of volatile organics in levitated viscous particles have shown strong
temperature dependence of bulk diffusivity and evaporation timescale (Bastelberger et
al., 2017). Slow bulk diffusion may impact multiphase processes such as browning of
organic particles (Liu et al., 2018), cloud droplet activation (Slade et al., 2017), and
ice nucleation pathways (Knopf et al., 2018).
Given these observations and strong implications of SOA phase states, it is
important to evaluate common assumption of gas-particle partitioning equilibrium at
different ambient conditions. In this study we provide theoretical analysis of
partitioning kinetics of organic compounds using the kinetic multi-layer model of
gas-particle interactions in aerosols and clouds (KM-GAP) (Shiraiwa et al., 2012),
which accounts for mass transport in both gas and particle phases. The equilibration
timescale ($\tau_{eq}$) of organic compounds partitioning into mono-dispersed particles is
evaluated systematically under a wide range of temperatures and RH, considering the
effects of particle phase state, particle size, mass loadings, and volatility of organic
compounds in a closed system with finite amount of vapor. For comparison we also
present simulations in an open system with vapor concentration maintained as
constant. This is the first study to directly relate equilibration timescale of SOA
partitioning to ambient temperature and relative humidity, which has important
implications in treatment of SOA evolution in chemical transport models.

**2. Methods**

We evaluate the timescale to achieve gas-particle equilibrium by simulating

condensation of a compound Z into pre-existing non-volatile mono-dispersed particles
using the KM-GAP model. KM-GAP consists of multiple model compartments and
layers, respectively: gas phase, near-surface gas phase, sorption layer, surface layer,
and a number of bulk layers (Shiraiwa et al., 2012). The following processes are
treated as temperature-dependent in KM-GAP: gas phase diffusion,
adsorption/desorption, surface-bulk exchange, and bulk diffusion (Fig. S1). The
physical and kinetic parameters are summarized in Table S1. The gas-phase diffusion
coefficient depends on temperature ($T$) and ambient pressure ($P$). $P$ is calculated as a
function of $T$ based on the International Standard Atmosphere
(https://www.iso.org/standard/7472.html). The adsorption rate coefficient is related to
the mean thermal velocity as a function of $T$ and the surface accommodation
coefficient, which is assumed to be 1 (Julin et al., 2014). The $T$-dependence of
desorption rate coefficient is described by an Arrhenius equation with an assumed
typical adsorption enthalpy of 40 kJ mol$^{-1}$.

Phase state and viscosity can be characterized by the glass transition

temperature ($T_g$), at which phase transition between amorphous solid and semi-solid
states occurs (Koop et al., 2011). When $T_g$ of organic particles under dry conditions
($T_{g,org}$) is known, $T_g$ of organic-water mixtures at given RH can be estimated
considering hygroscopic growth combined with the Gordon-Taylor equation. In this
work, we assumed the effective hygroscopicity parameter as 0.1 (Petters and
Kreidenweis, 2007; Gunthe et al., 2009) and the Gordon-Taylor constant as 2.5 (Koop
et al., 2011). Then, the $T$-dependence of viscosity is calculated using the
Vogel-Tammann-Fulcher equation (Angell, 1991; Rothfuss and Petters, 2017;
DeRieux et al., 2018; Li and Shiraiwa, 2018).

Figure 1 shows the $T$- and RH-dependent viscosity of SOA particles with $T_{g,org}$

of (a) 240 K, (b) 270 K, and (c) 300 K. We chose these three $T_{g,org}$ values to represent
different phase states of liquid, semi-solid, and glassy states, respectively, at $T$ of 298
K under dry conditions and these values are within the range recently reported for
monoterpene-derived SOA (Petters et al., 2019). The decrease of $T$ leads to increase
of viscosity, while the increase of RH leads to decrease of viscosity due to the
plasticizing effect of water (Koop et al., 2011). For simplicity we assume particles are
ideally-mixed, even though phase-separated particles are observed for ambient and
laboratory generated SOA particles under certain conditions (You et al., 2012;
Renbaum-Wolff et al., 2016). The bulk diffusion coefficient $D_b$ (Fig. S2) is calculated
by the Stokes–Einstein equation, which has been shown to work very well for organic
molecules diffusing through materials with viscosity below ~$10^3$ Pa s (Chenyakin et
al., 2017). Note that the Stokes–Einstein equation may underpredict $D_b$ in highly
viscous SOA thus it gives lower limits of $D_b$ (Price et al., 2015; Marshall et al., 2016;
Bastelberger et al., 2017; Reid et al., 2018). $D_b$ is fixed at any given depth in the
particle bulk in each simulation, assuming that condensation of Z would not alter
particle viscosity and diffusivity as only trace amounts of Z condense to pre-existing
particles in our simulations. Particle-phase reactions and their potential impacts on
particle visocisty are also not considered in this study.

We mainly consider a closed system, in which condensation of Z would lead

to a decrease of its gas-phase mass concentration ($C_g$) and an increase of its
particle-phase mass concentration ($C_p$). The particle diameter stays practically
constant throughout each simulation, as the amount of condesing Z is set to be much
smaller than the non-volatile pre-existing particle mass ($C_{OA}$). The gas-phase mass
concentration of Z right above the surface ($C_s$) is also calculated based on the Raoult's
law and partitioning theory (Pankow, 1994) in equilibrium with the near-surface bulk,
which is resolved by KM-GAP (Shiraiwa and Seinfeld, 2012). We also calculate the
mass fraction of Z in the near-surface bulk ($f_s$) and the average mass fraction of Z in
the entire bulk ($f_b$) to infer the radial concentration profile (Fig. S3). The equilibration
timescale ($\tau_{eq}$) is calculated as the e-folding time $t$ when the following criterion is met,
$$\frac{|C_p(t) - C_{p,eq}|}{|C_{p,0} - C_{p,eq}|} < \frac{1}{e} \qquad\qquad\qquad (1)$$

where $C_{p,0}$ and $C_{p,eq}$ are the initial and equilibrium mass concentration of Z in the
particle phase, respectively. Note that practically the same values can also be obtained
by using initial and equilibrium gas-phase concentrations in Eq. (1), as the mass
change of Z in the gas and particle phases are the same in these simulations.

**3 Results**
**3.1. Impacts of volatility and diffusivity on equilibration timescales**
Figure 2 shows exemplary simulations of temporal evolution of $C_g$ (blue line)
and $C_p$ (red line) of the compound Z in the closed system along with $\tau_{eq}$ marked with
red circles. The initial mass concentration of pre-existing non-volatile mono-dispersed
particles ($C_{OA}$) is assumed to be 20 μg m$^{-3}$ with the number concentrations of $3 \times 10^4$
cm$^{-3}$ and the initial particle diameter of 100 nm. Initial mass concentrations of Z in the
gas ($C_{g,0}$) and particle ($C_{p,0}$) phases are set to be 0.3 μg m$^{-3}$ and 0 μg m$^{-3}$, respectively.
$T_{g,org}$ is assumed to be 270 K. Figure 2a presents simulations for a semi-volatile
organic compound (SVOC) with the pure compound saturation mass concentration
($C^0$) of 10 μg m$^{-3}$ condensing on particles with $D_b$ of $10^{-11}$ cm$^2$ s$^{-1}$ at RH = 60% and $T$
= 298 K (Fig. S2). Upon condensation $C_g$ decreases, while $C_s$ and $C_p$ increase, and the
gas-particle equilibrium is reached within about 20 s as indicated by $\tau_{eq}$. For
low-volatile organic compounds (LVOC) with $C^0 = 0.1$ μg m$^{-3}$, it takes longer time to
reach the equilibrium with $\tau_{eq}$ of ~30 s (Fig. 2b), as the partial pressure gradient
between the gas phase and the particle surface (represented by the difference between
$C_g$ and $C_s$) is larger for lower $C^0$. For both cases SOA growth is governed by
gas-phase diffusion as indicated by $C_s < C_g$. The mass fraction of Z in the near-surface
bulk is identical to the average mass fraction in the entire bulk (Fig. S3 a–b),
indicating that Z are homogeneously well-mixed in the particle without kinetic
limitations of bulk diffusion in low viscous particles (Fig. 3a).
At lower $T$ of 250 K, the phase state of pre-existing particles occurs as highly
viscous with $D_b$ of ~$10^{-18}$ cm$^2$ s$^{-1}$ (Fig. S2), resulting in much longer equilibration
timescales (~$10^5$ s) for SVOC with $C^0$ = 10 μg m$^{-3}$ (Fig. 2c). After $C_g$ and $C_s$
converge, they continue to decrease simultaneously while $C_p$ increases slowly,
showing that the particle undergoes quasi-equilibrium growth (Shiraiwa and Seinfeld,
2012; Zhang et al., 2012). For LVOC ($C^0$ = 0.1 μg m$^{-3}$) condensation, $\tau_{eq}$ is short
(~140 s) because of a local thermodynamic equilibrium between the gas phase and the
near-surface bulk established relatively quickly (as mostly controlled by the
condensation sink: Riipinen et al., 2011; Tröstl et al., 2016) due to very slow
re-evaporation of LVOC.
The characteristic timescale of mass transport and mixing by molecular
diffusion $\tau_{mix}$ can be calculated by $\tau_{mix} = r_p^2 / (\pi^2 D_b)$, where $r_p$ is the particle radius
(Seinfeld and Pandis, 2006). Figure 3 shows dimensionless radial concentration
profiles of Z ($C^0$ = 0.1 μg m$^{-3}$) in the particle at (a) $D_b = 10^{-11}$ cm$^2$ s$^{-1}$ and (b) $10^{-18}$ cm$^2$
s$^{-1}$, respectively. For low viscous particles, $\tau_{mix}$ is very short and particles are
homogeneously well-mixed at $\tau_{eq}$, which is consistent with previous analytical
calculations (Liu et al., 2013; Mai et al., 2015). In contrast, there exists a large
concentration gradient between the particle surface and the inner bulk (Fig. 3b, S3d)
at $\tau_{eq}$ in highly viscous particles due to strong kinetic limitations of bulk diffusion (as
indicated by very long $\tau_{mix}$), which prevents the entire particle bulk to reach complete
equilibrium. Thus, for LVOC condensation on highly viscous particles (Fig. 2d), $\tau_{mix}$
represents the timescale to establish full equilibrium with homogeneous mixing in the
entire particle bulk. These results are consistent with Mai et al. (2015) and Liu et al.
(2016), which showed that an establishment of full equilibrium is limited by bulk
diffusion in highly viscous particles, even though the local equilibrium of LVOC may
be achieved faster. Note that $\tau_{mix}$ is solely a function of particle size and bulk
diffusivity, while $\tau_{eq}$ is also affected by volatility and mass loadings. At lower particle
concentrations, the total accommodation of molecules to the particle surface
decreases, resulting in longer equilibration timescales (Fig. S4).

We further computed $\tau_{eq}$ as a function of $D_b$ and $C^0$ in the closed system. As

shown in Fig. 4a, when $D_b$ is higher than $\sim 10^{-13}$ cm$^2$ s$^{-1}$, $\tau_{eq}$ is insensitive to bulk
diffusivity but sensitive to volatility: decreasing volatility increases $\tau_{eq}$ in this regime.
In the regime with $D_b$ lower than $\sim 10^{-13}$ cm$^2$ s$^{-1}$ and $C^0$ higher than $\sim 10$ μg m$^{-3}$, $\tau_{eq}$ is
controlled by bulk diffusivity: $\tau_{eq}$ increases from 30 s to longer than 1 year as $D_b$
decreases from $10^{-13}$ cm$^2$ s$^{-1}$ to $10^{-20}$ cm$^2$ s$^{-1}$. In the regime with $D_b < \sim 10^{-13}$ cm$^2$ s$^{-1}$
and $C^0 < \sim 10$ μg m$^{-3}$, $\tau_{eq}$ depends on both diffusivity and volatility. Decreasing
volatility would lead to shorter $\tau_{eq}$ due to an establishment of local equilibrium of
LVOC.
In an open system with fixed vapor concentration (Fig. S5), $\tau_{eq}$ of SVOC is
slightly longer but on the same order of magnitude as $\tau_{eq}$ in the closed system, as
relatively small amounts of SVOC need to condense to reach equilibrium. In contrast,
$\tau_{eq}$ of LVOC in the open system become dramatically longer as LVOC continue to
condense into the particle phase because of low volatility (Pankow, 1994). For further
simulations we focus mainly on the closed system and the corresponding simulations
for the open system are provided in the supplement.
We also simulated evaporation in the closed system with same parameters as
the condensation simulations (Table S2). Initially $C_g = 0$ µg m$^{-3}$ and trace amounts of
semi-volatile or low-volatile species were assumed to be homogeneously well-mixed
in pre-existing particles. Figure S6 shows that for the evaporation of SVOC species
with $C^0 = 10$ µg m$^{-3}$, decreasing $D_b$ from $10^{-11}$ cm$^2$ s$^{-1}$ to $10^{-18}$ cm$^2$ s$^{-1}$ would increase
$\tau_{eq}$ from $\sim 20$ s to $\sim 10^5$ s. These evaporation timescales are close to those derived
from condensation (Fig. 2a,c) and consistent with previous kinetic simulations (Liu et
al., 2016). In the closed system, the evaporation of a very small amount of LVOC
species from the particle surface is already sufficient to reach the particle-phase
equilibrium concentration, resulting in a short $\tau_{eq}$ (Fig. S6b,d). For an open system
with continuous removal of gas-phase compounds, which is often employed in
evaporation experiments, the equilibrium timescale in the evaporation of the LVOC
species from highly viscous particles can be longer than hours or days (Vaden et al.,
2011; Liu et al., 2016). Figure 4b shows simulated evaporation timescales as a
function of $D_b$ and $C^0$ in an open system, which agrees very well with Fig. 3 in Liu et
al. (2016). It shows that for less viscous particles $\tau_{eq}$ is limited by volatility, while for
highly viscous particles $\tau_{eq}$ is insensitive to volatility and controlled by bulk
diffusivity.

**3.2. Equilibration timescales at different RH and *T***

We conducted further simulations to estimate $\tau_{eq}$ with a wide range of
atmospherically-relevant temperatures (220 - 310 K) and relative humidities (0 -
100%). Figure 5 shows the temperature and humidity-dependent diagrams of $\tau_{eq}$ for
SVOC ($C^0 = 10$ μg m$^{-3}$) condensation on particles with $T_{g,org}$ of 240 K, 270 K, and
300 K, respectively, in the closed system. For particles with $T_{g,org}$ of 240 K (panel a),
$\tau_{eq}$ is on the order of seconds under boundary layer conditions ($T > 270$ K). In these
conditions particles are liquid with high bulk diffusivity (Fig. 1a and S2a), thus
gas-particle partitioning is controlled by gas-phase diffusion and interfacial transport
(Shiraiwa and Seinfeld, 2012; Mai et al., 2015). At low $T$ (< 260 K) with low or
moderate RH (< 70%), $\tau_{eq}$ can increase from minutes to one year with decreasing $T$
and RH mainly due to strong kinetic limitations of bulk diffusion with low $D_b$ (Fig.
S2a). With $T_{g,org}$ of 270 K (panel b) or 300 K (panel c), $\tau_{eq}$ is still on the order of
minutes in most of boundary layer conditions. At low RH $\tau_{eq}$ can be extended to hours
when particles may occur as amorphous (semi-)solid. When $T < 270$ K, $\tau_{eq}$ can be
longer than months even at moderate RH, while $\tau_{eq}$ may stay very short at very high
RH. The corresponding simulations of SVOC partitioning in the open system (Fig.
S7) show a similar pattern as $\tau_{eq}$ in the closed system.

$\tau_{eq}$ for $C^0 = 10^3$ and 0.1 μg m$^{-3}$ in the closed system are presented in Fig. A1. In

general, $\tau_{eq}$ would be shorter at higher $C^0$ when particles are liquid, as the partial
pressure gradient between the gas phase and the particle surface is smaller for higher
$C^0$ (Shiraiwa and Seinfeld, 2012; Liu et al., 2016). For example, the increase of $C^0$
from 10 μg m$^{-3}$ to $10^3$ μg m$^{-3}$ leads to $\tau_{eq}$ decrease from 30 s to 1 s with $T_{g,org}$ of 240 K
at boundary layer conditions (Fig. 5a, A1a). At low $T$ and RH (e.g., $T < 250$ K and
RH < 50 %) where particles are highly viscous, $\tau_{eq}$ is on the same order of magnitude
for the condensation of IVOC and SVOC, as gas-particle partitioning is limited by
bulk diffusion. Figure A2 shows bulk diffusion and mixing timescales ($\tau_{mix}$) as a
function of RH and $T$. It is interesting to note that $\tau_{mix}$ is very similar to $\tau_{eq}$ of IVOC
(Fig. A1(a-c)) as gas diffusion and interfacial transport of IVOC are fast. For LVOC
$\tau_{eq}$ is generally shorter than $\tau_{mix}$ as its mass transfer to the particle surface is governed
by condensation sink with negligible re-evaporation, while $\tau_{mix}$ is still long to achieve
homogeneous mixing in the particle phase if particles are viscous.

Previous studies have shown that $\tau_{eq}$ depends on particle size (Liu et al., 2013;

Zaveri et al., 2014; Mai et al., 2015) and particle mass loadings (Shiraiwa and
Seinfeld, 2012; Saleh et al., 2013). For further examination of these effects at
different $T$, Figure 6 shows the dependence of $\tau_{eq}$ of SVOC ($C^0 = 10$ μg m$^{-3}$) and
LVOC ($C^0 = 0.1$ μg m$^{-3}$) on the mass concentration and the diameter of pre-existing
particles, over the range of 0.1 – 100 μg m$^{-3}$ and 30 – 1000 nm, respectively, with
particle phase state to be less viscous with $D_b = 10^{-11}$ cm$^2$ s$^{-1}$ at 298 K and highly
viscous with $D_b = 10^{-18}$ cm$^2$ s$^{-1}$ at 250 K. In this comparison, when ambient particle
mass concentration is held constant, increasing particle size will translate to a
decrease of the number and surface area concentration of particles, and a decrease of
total accommodation of molecules to the particle surface, thereby leading to an
increase of $\tau_{eq}$. When particle diameter is held constant, an increase of particle
concentration leads to an increase of surface area concentration, resulting in shorter
$\tau_{eq}$. When particles are less viscous at 298 K ($D_b = 10^{-11}$ cm$^2$ s$^{-1}$) $\tau_{eq}$ of SVOC is
shorter than that of LVOC for the same particle size and mass loadings. For
partitioning into highly viscous particles at 250 K ($D_b = 10^{-18}$ cm$^2$ s$^{-1}$), SVOC takes
longer time than LVOC to reach equilibrium.

Typical ambient organic mass concentrations in Beijing, Centreville in

southeastern US, Amazon Basin, and Hyytiälä, Finland are indicated in Fig. 6. The
particle phase state was observed to be mostly liquid in highly polluted episodes in
Beijing (Liu et al., 2017), under typical atmospheric conditions in the southeastern US
(Pajunoja et al., 2016), and under background conditions in Amazonia (Bateman et
al., 2017). At these conditions $\tau_{eq}$ should be mostly less than 30 minutes (Fig. 6a, b).
Particles were measured to be semi-solid or amorphous solid in clear days in Beijing
(Liu et al., 2017), in Amazonia when influenced by anthropogenic emissions
(Bateman et al., 2017), and the boreal forest in Finland (Virtanen et a., 2010). Under
these conditions and also when particles are transported to the free troposphere, $\tau_{eq}$
can be longer than 1 hour especially in remote areas with low mass loadings (Fig. 6c,
d). Particles in nucleation mode (diameter < 30 nm) are not considered in this study,
as the particle size may affect the phase transition of these nanoparticles (Cheng et al.,
2015). The role and impact of phase transition on nucleation and growth of ultrafine
particles are beyond the scope of current simulations and need further investigations
in future studies.

**4 Discussion**

The timescale to reach equilibrium for SOA partitioning has been investigated

in several laboratory experiments at room temperatures (Vaden et al., 2011; Saleh et
al., 2013; Liu et al., 2016; Ye et al., 2016a; Gong et al., 2018; Ye et al., 2018). These
experiments monitored particle mass or composition, finding that equilibration
timescales are longer at low RH, consistent with our model simulations. Note that, for
condensation on highly viscous particles, even though particle mass or particle-phase
concentrations appear to reach equilibrium, complete equilibrium with homogeneous
mixing in the particle may not have been reached driven by strong kinetic limitations
and concentration gradients in the particle bulk (Fig. 2d and 3b). This is also
supported by evaporation experiments showing that the local thermodynamic
equilibrium established between the vapor and the near-surface bulk should be
differentiated from the global equilibrium between the vapor and the entire bulk (Liu
et al., 2016). Note that SOA evaporation is also influenced by volatility and oligomer
decomposition (Roldin et al., 2014; Yli-Juuti et al., 2017). The timescale of
gas-particle partitioning can be different in closed or open systems especially for
LVOC (Fig. 4, S7). The closed system simulations represent SOA partitioning in
chamber experiments and in closed atmospheric air mass, which could be justified
well within seconds-to-minutes timescales and possibly up to hours depending on
meteorological conditions. The real atmosphere may be approximated better as an
open system due to dilution and chemical production and loss especially at longer
timescales. Thus, particular care needs to be taken in comparing modeling results with
different field conditions or experiments on probing equilibration timescale (i.e.,
evaporation vs. condensation, open vs. closed system, local vs. full equilibrium).
The simulated equilibration timescales of atmospheric SOA are mostly on the
order of minutes to hours under conditions of atmospheric boundary layer (Fig. 5,
A1). This agrees with previous experimental results that the gas-particle interactions
can be regulated by both thermodynamic and kinetic partitioning (Booth et al., 2014;
Liu et al., 2016; Saha and Grieshop, 2016; Ye et al., 2016a; Gong et al., 2018),
depending on several factors including particle phase state, size, mass loadings, and
volatility. Organic particles containing high molar-mass compounds tend to have high
glass transition temperatures (Koop et al., 2011) and the occurrence of kinetic
limitation will increase with higher $T_{g,org}$ (Fig. 5). This is consistent with the results of
intraparticle mixing experiments showing that as the carbon number of precursor (e.g.
terpene) increased (that would lead to higher $T_{g,org}$), it took longer time for SVOCs
(evaporated from another type of SOA, e.g. toluene SOA) to partition into the terpene
SOA, leading to slower molecular exchange among different types of SOA (Ye et al.,

2018).

At low temperatures, the particles can occur as highly viscous at relatively

high RH (Fig. 1), and $\tau_{eq}$ of SVOC partitioning can be longer than hours or days (Fig.
5, S7). Equilibration timescales of LVOC condensation at low particle mass loadings
(Fig. 6) may represent the clean conditions where new particle formation and growth
often occur (Wang et al., 2016). It has been reported that highly oxygenated
molecules play an important role in the initial growth of atmospheric particles in the
free troposphere (Bianchi et al., 2016). Bulk diffusion would likely to be a limiting
step in the condensation of semi-volatile and low volatility compounds at low
temperatures, where particles may occur as highly viscous (Shiraiwa et al., 2017). In
this case, particle growth would need to be treated kinetically, rather than
thermodynamic equilibrium partitioning, as it would affect SOA growth kinetics and
size distribution dynamics, with significant implications for the growth of ultrafine
particles to climatically relevant sizes (Riipinen et al., 2011; Riipinen et al., 2012;
Shiraiwa et al., 2013a; Zaveri et al., 2018). Chemical transport models usually have
time steps on the order of minutes, within which the partitioning equilibrium may not
be reached, for most SVOC species ($C^0 > 1$ μg m$^{-3}$) when $D_b$ is less than $10^{-15}$ cm$^2$ s$^{-1}$
(Fig. 4). Note that condensation of extremely low volatility organic compounds
(ELVOC; Tröstl et al., 2016) into highly viscous particles may be governed by
gas-phase diffusion and timescales to reach local equilibrium could be shorter as
determined by the condensation sink (Riipinen et al., 2011) (see also Fig. S4b), which
may be more relevant for the practical application in chemical transport models.

In this study we assume that the bulk diffusivity within organic particles is

independent of particle mixing state and morphology. Chamber experiments have
demonstrated that evaporation of organic aerosol may be hindered if it is coated with
organic aerosol from a different precursor (Loza et al., 2013; Boyd et al., 2017).
Moreover, the phase separation has been observed in organic particles mixed with
inorganic salts (You et al., 2014) and even without inorganic salts (Pöhlker et al.,
2012; Riedel et al., 2016). Future simulations on equilibration timescale should
consider the effects of the immiscibility (Barsanti et al., 2017; Liu et al., 2013) and
the phase separation (Shiraiwa et al., 2013b; Pye et al., 2017; Fowler et al., 2018) as
well as composition-dependent bulk diffusivity (O'Meara et al., 2016) and the
evolution of the particle phase due to reactive uptake and condensed-phase chemistry
(Hosny et al., 2016). Incorporation of the particle-phase formation of oligomers and
other multifunctional high molar mass compounds can lead to a reduced bulk
diffusivity (Pfrang et al., 2011; Hosny et al., 2016), which may prolong the
equilibration timescales. Decomposition of highly oxidized molecules (e.g., organic
hydroperoxides) in water may also affect gas-particle partitioning (Tong et al., 2016).
Current simulations are focused on trace amount of SVOC or LVOC condensing on
mono-dispersed particles with negligible particle growth. Potential phase transition in
the course of particle growth/evaporation should also be incorporated in future
simulations. The shift in particle phase state and gas-particle partitioning in response
to temperature and RH may need to be considered in chemical transport models and
laboratory experiments to better understand the fate of organic compounds.

**Author contribution.**

YL and MS designed and conducted modeling and wrote the manuscript.


**Acknowledgments.**

This work was funded by the National Science Foundation (AGS-1654104) and the Department of Energy (DE-SC0018349). The simulation data may be obtained from the corresponding author upon request.

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

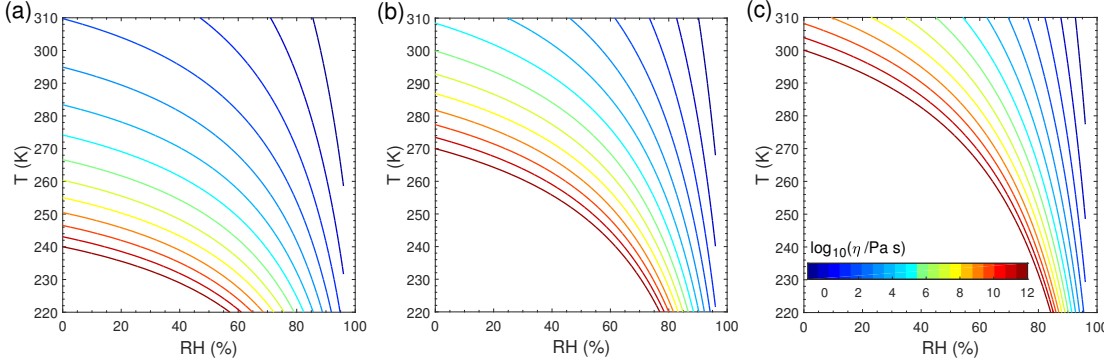

**Figure 1.** Viscosity of pre-existing particles as a function of temperature and relative humidity. The glass transition temperatures under dry conditions ($T_{g,org}$) are (a) 240 K, (b) 270 K, and (c) 300 K, respectively.

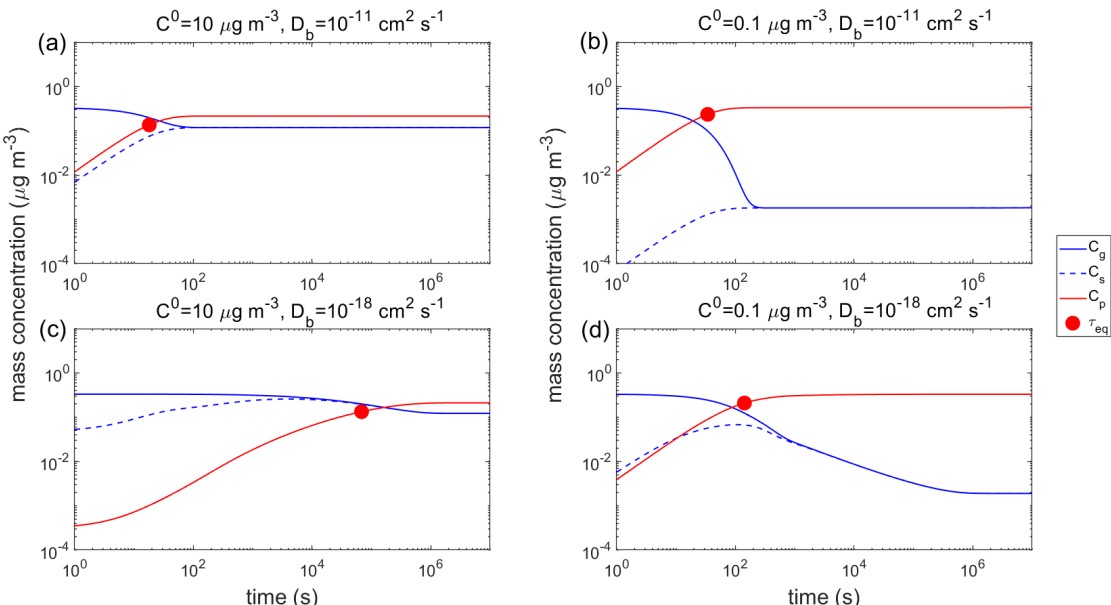

**Figure 2.** Temporal evolution of mass concentrations of the condensing compound Z in the gas phase ($C_g$), just above the particle surface ($C_s$), and in the particle phase ($C_p$) in the closed system. $\tau_{eq}$ is marked with the red circle. RH = 60% and $T$ is (a–b) 298 K and (c–d) 250 K. The $C^0$ of Z is (a, c) 10 μg m$^{-3}$ and (b, d) 0.1 μg m$^{-3}$. The glass transition temperature of pre-existing particles under dry conditions ($T_{g,org}$) is set to be 270 K, which leads to $D_b$ of (a–b) 10$^{-11}$ cm$^2$ s$^{-1}$ and (c–d) 10$^{-18}$ cm$^2$ s$^{-1}$. The initial mass concentration of pre-existing particles is set to be 20 μg m$^{-3}$ with the number concentrations of $3 \times 10^4$ cm$^{-3}$ and the initial particle diameter of 100 nm.

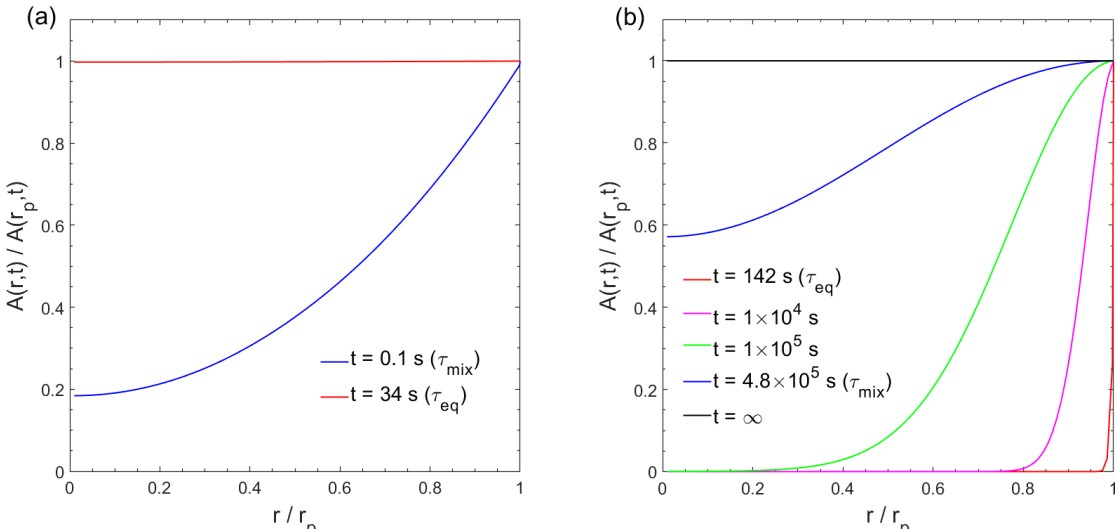


**Figure 3.** Dimensionless radial concentration profiles in the particle for the
condensation of the LVOC species ($C^0 = 0.1\ \mu g\ m^{-3}$) at RH = 60% and (a) $T$ = 298 K
with $D_b = 10^{-11}\ cm^2\ s^{-1}$ and (b) $T$ = 250 K with $D_b = 10^{-18}\ cm^2\ s^{-1}$. The x-axis indicates
the radial distance from the particle center ($r$) normalized by the particle radius ($r_p$),
ranging from the particle core ($r / r_p \approx 0$) to the surface ($r / r_p$ =1). The y-axis indicates
the bulk concentration of the condensing compound at a given position in the bulk ($r$)
normalized by the bulk concentration at particle surface ($r_p$).

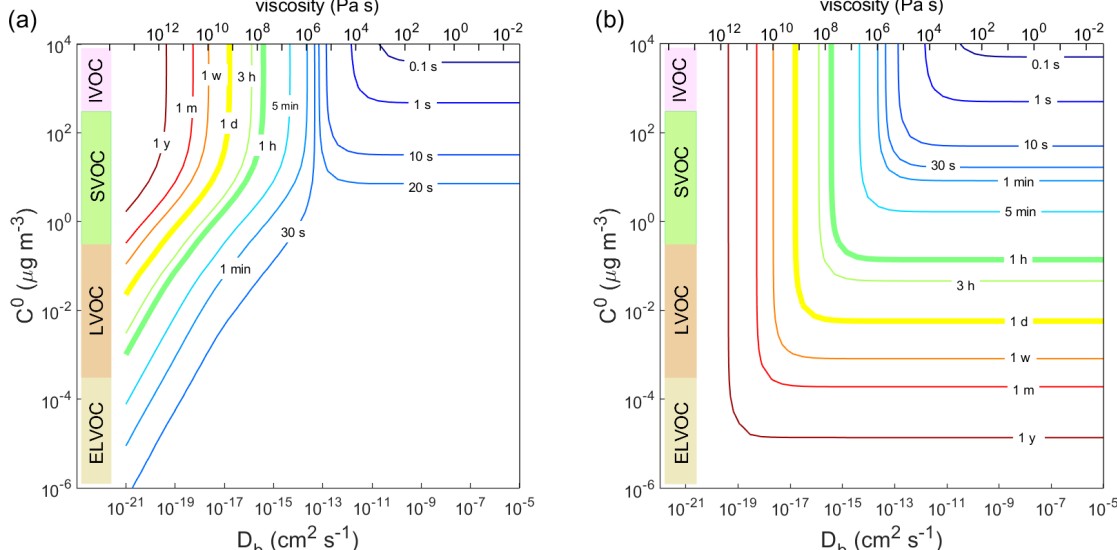

**Figure 4.** Contour plot of equilibration timescale ($\tau_{eq}$) as a function of bulk diffusivity ($D_b$) and saturation mass concentration ($C^0$) for (a) condensation in the closed system and (b) evaporation in the open system. The initial mass concentration of pre-existing particles is set to be 20 μg m$^{-3}$ with the number concentrations of $3 \times 10^4$ cm$^{-3}$ and the initial particle diameter of 100 nm. Viscosity is calculated from the Stokes-Einstein equation assuming the effective molecular radius of $10^{-8}$ cm at $T$ of 298 K.

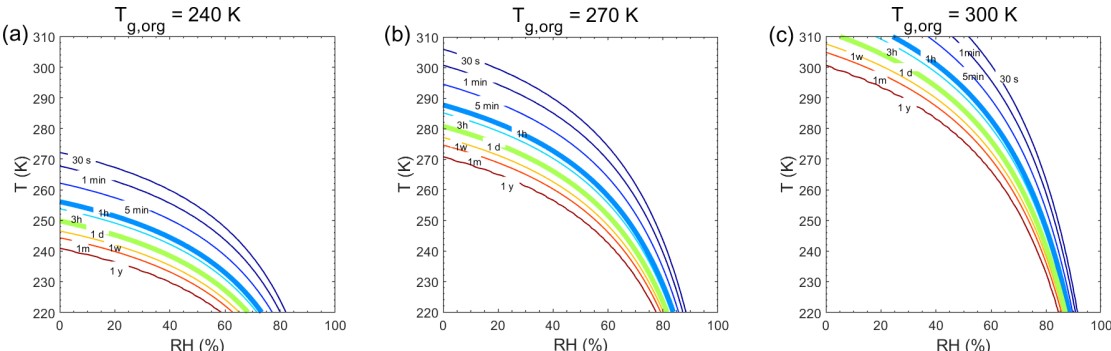

**Figure 5.** Equilibration timescale ($\tau_{eq}$) as a function of temperature and relative humidity in the closed system. The glass transition temperatures of pre-existing particles at dry conditions ($T_{g,org}$) are (a) 240 K, (b) 270 K, and (c) 300 K, respectively. The saturation mass concentration ($C^0$) of the condensing compound is 10 μg m$^{-3}$ (SVOC). The mass concentration of pre-existing particles is set to be 20 μg m$^{-3}$ with the number concentrations of $3 \times 10^4$ cm$^{-3}$ and the initial particle diameter of 100 nm.

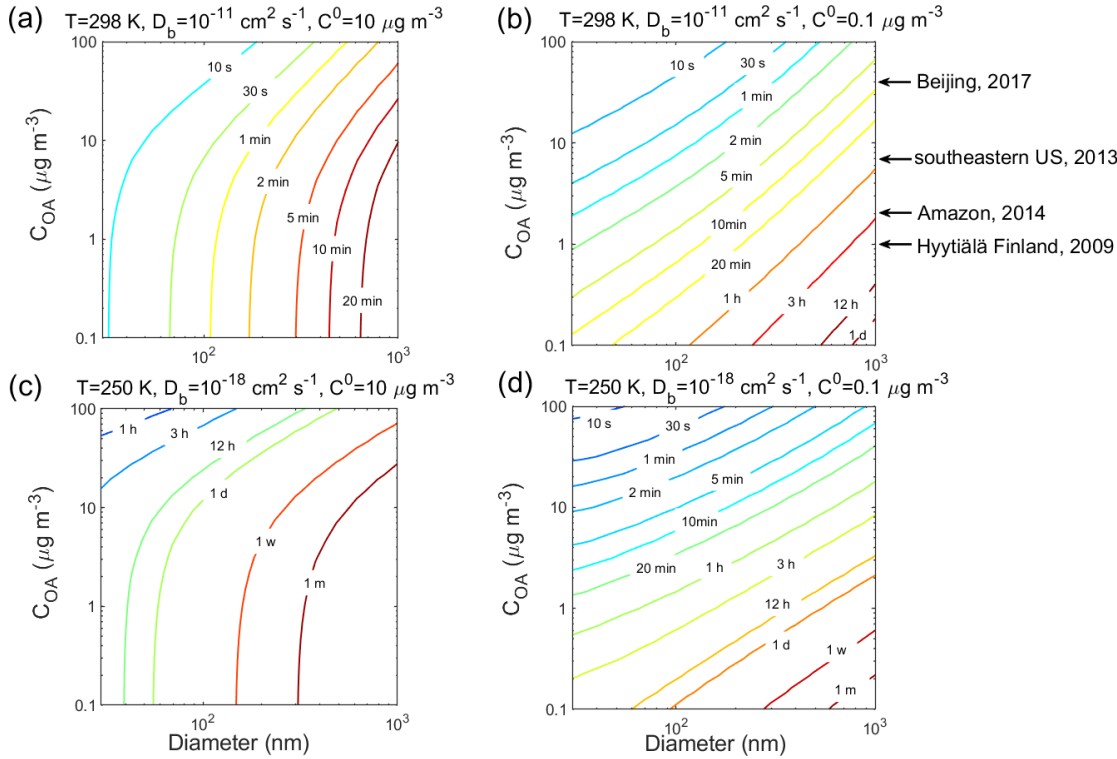

**Figure 6.** Equilibration timescale ($\tau_{eq}$) for (a, c) SVOC ($C^0 = 10$ μg m$^{-3}$) and (b, d) LVOC ($C^0 = 0.1$ μg m$^{-3}$) as a function of particle diameter (nm) and mass concentration (μg m$^{-3}$) of pre-existing particles at 60% RH and $T$ of (a-b) 298 K and (c-d) 250 K in the closed system. The glass transition temperature of pre-existing particles under dry conditions ($T_{g,org}$) is set to be 270 K, which leads to $D_b$ of (a-b) 10$^{-11}$ cm$^2$ s$^{-1}$ and (c-d) 10$^{-18}$ cm$^2$ s$^{-1}$. Ambient organic mass concentrations are indicated with arrows.

 **Appendix:**

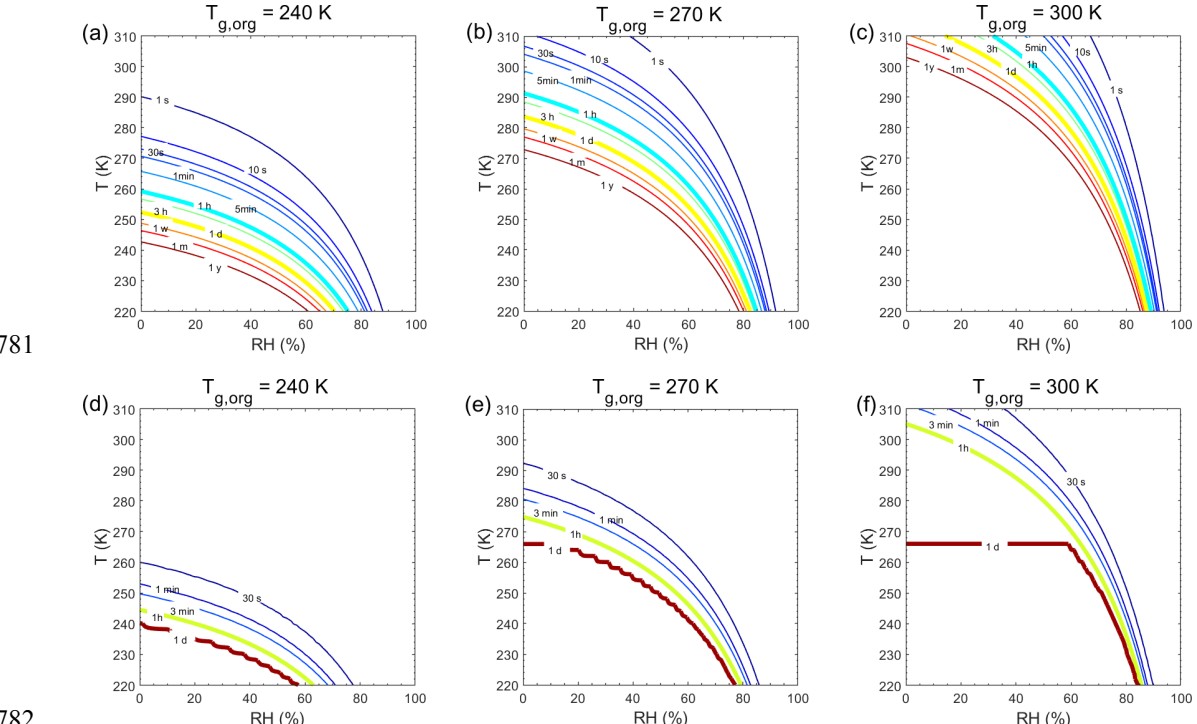

**Figure A1.** Equilibration timescale ($\tau_{\mathrm{eq}}$) as a function of temperature and relative humidity in the closed system. The glass transition temperatures of pre-existing particles at dry conditions ($T_{\mathrm{g,org}}$) are set to be (a, d) 240 K, (b, e) 270 K, and (c, f) 300 K. The mass concentration of pre-existing particles is 20 µg m$^{-3}$. The saturation mass concentration ($C^{0}$) of the condensing compound is (a, b, c) 10$^{3}$ µg m$^{-3}$ and (d, e, f) 0.1 µg m$^{-3}$.

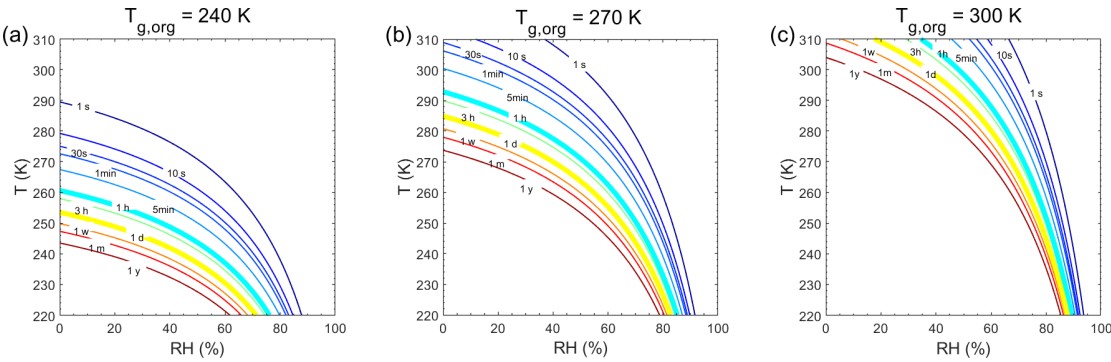

**Figure A2.** Characteristic timescale of bulk diffusion or mixing timescale ($\tau_{\mathrm{mix}}$) as a function of temperature and relative humidity. The particle diameter is assumed to be 100 nm with the glass transition temperatures of pre-existing particles at dry conditions ($T_{\mathrm{g,org}}$) of (a) 240 K, (b), 270 K and (c) 300 K.