# Peer review of "Timescales of Secondary Organic Aerosols to Reach Equilibrium at Various Temperatures and Relative Humidities"

_Atmospheric Chemistry and Physics, 2019_

## Referee Comment (RC1) · Anonymous Referee #1 · 7 Mar 2019

This manuscript explores the equilibrium timescale and mixing timescale of IVOC and LVOC with particles considering different phase states. The work combines the authors' previous KM-GAP model (Shiraiwa et al., 2012) with the authors' recent glass transition model (Shiraiwa et al., 2017; DeRieux et al. 2018) to understand the interplay among equilibrium timescale, temperature, relative humidity, and the glass transition temperature of the aerosols.

Besides the numerical results obtained from the model shown in Figure 1-5, the manuscript provides two more useful results: (1) when there is no diffusion limitation within the particle, the gases that have higher saturation mass concentrations will

reach gas-particle equilibrium faster (2) when there is strong diffusion limitation within the particle, gases that have lower volatility may reach gas-particle equilibrium (locally) faster than VOCs with higher volatility. One of the implications is that at a lower temperature (upper troposphere) or when dealing with highly viscous particles, the particle growth process may need to be treated kinetically.

The authors used a numerical model to obtain result #1 above, and it agrees with the findings in Liu et al. 2012 in which an analytical model was used. It is nice to see two different approaches come with the same results and can validate each other. On the other hand, result #2 is more interesting because it shows that local equilibrium could be reached relatively faster for low volatility species when the particle phase is highly viscous. The manuscript also illustrates some concepts that can be commonly misused by folks, such as the difference between equilibrium timescale and the mixing timescale.

Because some of the results have been previously discussed in or can be easily inferred from other publications (Liu et al. 2012, Shiraiwa et al. 2011&2012), the novelty of the manuscript needs to be improved. I suggest that the author focus on result #2, which is novel, and use it to make further predictions regarding the physical and chemical processes of aerosols. For instance, what is the relationship between particle sizes and condensation/evaporation kinetics of gases with different volatility interacting with particles with various viscosity? I would also be curious to know what is the boundary criteria for result #2 to happen, i.e., how viscous would the particles have to be or how low the volatility of the VOC have to be in order to reach relatively fast local equilibrium? The author can also explore the effects of equilibrium partitioning when the gases can both partition in and react with the particle phase.

My other question is that most of the modeling results shown the manuscript assumed that the gas-particle is in a closed system. How realistic is the closed system in ambient environment? Would the ambient environment often be an open system for evaporation kinetics?

[Figure]

The author should also include Price et al. 2015 in the reference list in line 115.

Reference: Shiraiwa et al. 2011: doi.org/10.1073/pnas.1103045108 Liu et al. 2012: doi.org/10.1080/02786826.2012.730163 Shiraiwa et al. 2012: doi.org/10.1029/2012GL054008 Price et al. 2015: doi.org/10.1039/C5SC00685F
* * *

---

## Referee Comment (RC2) · Anonymous Referee #2 · 9 Mar 2019

Review of Li and Shiraiwa "Timescales of Secondary Organic Aerosol to Reach Equilibrium at Various Temperatures and Relative Humidities"

**General Comments**

In this manuscript, the authors use simulations to calculate the gas/particle equilibration time for secondary organic aerosol as a function of temperature, relative humidity, and SOA microphysical properties. The topic of SOA partitioning, equilibration, and phase state are highly relevant in atmospheric chemistry, the topic is timely, and will be of interest to the readers of ACP.

This work is a logical extension of the previous work done by the PI. It essentially combines work in predicting the phase state of particles as a function of atmospheric conditions with work on calculating gas/particle equilibration times (Shiraiwa et al., 2017;Shiraiwa and Seinfeld, 2012). Based on past results from the PI on these topics, the present results are not particularly surprising, but I think there is enough new material here to warrant publication. This publication essentially closes the loop between predicting phase state and calculating the gas/particle equilibration time.

With that said, there are a few areas that could be improved before publication. The authors could do a better job of calling out, in the manuscript, what is new unique about this manuscript relative to the previous publications by the PI. A few of the conclusions reached in this work also seem to contradict the PI's previous publications and the authors should clear this up. The authors could help readers put this work into context if they explain why they make certain assumptions in their model (i.e., a closed system), offer some insights into how realistic these assumptions are relative to the ambient atmosphere, and explain how their conclusions would be different if/when the assumptions are not atmospherically representative. While the manuscript is generally well written and clear, there was one section that was somewhat confusing and should be clarified before publication. With that said, there are no major shortcomings with the manuscript and, providing the authors make some revisions, I have no reservations about recommending this manuscript for publication in ACP.

**Major Specific Comments**

**Lines 23-25, Figure A1, and elsewhere.** In the present manuscript the authors conclude that the equilibration timescale for low volatile compounds is shorter than for semi-volatile compounds when other conditions are equal. However, Shiraiwa and Seinfeld 2012 report the opposite (see for example Figure 2), with ELVOC's having longer equilibration times than SVOCs (Shiraiwa and Seinfeld, 2012). The authors should comment in the text on why these studies reach opposite conclusions.

**Lines 115-123.** I understand that the authors need to make some assumptions or approximations in their model/calculations. I am trying to understand how atmospherically realistic these

assumptions are. The two main assumptions in the present model are of a closed system and that the condensation of molecule Z does not alter the composition and microphysical properties of the pre-existing particles.

It is clear that the real atmosphere isn't a closed system. The argument could be made that on seconds-to-minutes timescales, it may approximate a closed system, but the processes that authors are modeling are sometimes occurring on timescales of hours or even days. In addition, one of the author's major conclusions is that low volatility material reaches equilibrium more slowly than higher volatility material. I understand why this is the case in a closed system, but would this conclusion also hold in an open system like in the atmosphere with a constant dilution and/or loss of gas-phase molecules? A plume transported from the surface to the upper troposphere would experience an evaporative driving force where this model seems focused on cases where the driving force is toward the particle phase (condensation).

With respect to a single compound (Z) changing the composition and microphysical properties, it may be true that a single molecule or even a few molecules rarely make up the bulk of the SOA mass. However, in the real atmosphere, particles obviously grow and their composition and microphysical properties change as SOA condenses. The PI of this manuscript has previously used kinetic modeling to reproduce particle growth, so I'm not really clear on why these assumptions needed to be made (Shiraiwa et al., 2013).

In the both cases, I think it is important that the authors explain: why they chose to make these assumptions, how likely it is that these assumptions are representative of the atmosphere, and how their conclusions would likely be different if the assumptions are not correct.

**Lines 276-292 and Figure 5**. This figure and associated text was confusing. The figure was confusing because particle diameter and total mass loading are typically not independent of one another in a model or in the atmosphere. I eventually understood the point the authors were trying to make. Perhaps providing some context by pointing out where different atmospheric regimes lie (i.e., remote, typical continental, polluted) in the figure would help. The associated text is also confusing; it wasn't clear what point the authors were trying to make here. They seem to postulate several different processes which could determine particle equilibration timescales (e.g., bulk diffusion, gas-diffusion). Can't the KM-GAP model be used to clear this up? Overall, I'm not sure what message the authors are trying to convey here.

**Minor Comments and Technical Corrections**

Line 89. It is a little confusing here about what temperature was used in the calculations. I wasn't clear whether 273.15 K was used (the most common definition of standard temperature), or if the temperature was variable as a function of pressure altitude. Adding to the confusion, Table S2 lists standard temperature as 288.15 K. Please clarify.

Lines 108-110. An assumption here is that the organic and aqueous phases are not phase separated. The authors point out later in the manuscript that phase separations may occur, but I suggest also briefly mentioning that phase separation has been observed for laboratory generated SOA (You et al., 2012) here, since it is very relevant to their modeling results.

Lines 144 and elsewhere. I found the use of $C_0$ as an abbreviation more confusing than necessary. It seems $C_0$ is identical to the much more commonly used $C^*$. Why not use the commonly accepted $C^*$? The authors also use $C_{p,0}$ and $C_{g,0}$, which have a different meaning and cause some confusion with $C_0$. Whatever symbol the author use for the saturation vapor pressure please define it the first time it is used.

Figures 4, A1, A2. The labels on different contours were illegible on the printed document.

**References**

Shiraiwa, M., and Seinfeld, J. H.: Equilibration timescale of atmospheric secondary organic aerosol partitioning, Geophysical Research Letters, 39, doi:10.1029/2012GL054008, 2012.

Shiraiwa, M., Yee, L. D., Schilling, K. A., Loza, C. L., Craven, J. S., Zuend, A., Ziemann, P. J., and Seinfeld, J. H.: Size distribution dynamics reveal particle-phase chemistry in organic aerosol formation, Proceedings of the National Academy of Sciences of the United States of America, 110, 11746-11750, 10.1073/pnas.1307501110, 2013.

Shiraiwa, M., Li, Y., Tsimpidi, A. P., Karydis, V. A., Berkemeier, T., Pandis, S. N., Lelieveld, J., Koop, T., and Pöschl, U.: Global distribution of particle phase state in atmospheric secondary organic aerosols, Nature Communications, 8, 15002, 10.1038/ncomms15002 https://www.nature.com/articles/ncomms15002#supplementary-information, 2017.

You, Y., Renbaum-Wolff, L., Carreras-Sospedra, M., Hanna, S. J., Hiranuma, N., Kamal, S., Smith, M. L., Zhang, X., Weber, R. J., Shilling, J. E., Dabdub, D., Martin, S. T., and Bertram, A. K.: Images reveal that atmospheric particles can undergo liquid-liquid phase separations, Proceedings of the National Academy of Sciences of the United States of America, 109, 13188-13193, 10.1073/pnas.1206414109, 2012.

---

## Author Comment (AC1) · 5 Apr 2019

**Response to the comments of Anonymous Referee #1**

Referee General Comment:

This manuscript explores the equilibrium timescale and mixing timescale of IVOC and LVOC with particles considering different phase states. The work combines the authors' previous KM-GAP model (Shiraiwa et al., 2012) with the authors' recent glass transition model (Shiraiwa et al., 2017; DeRieux et al. 2018) to understand the interplay among equilibrium timescale, temperature, relative humidity, and the glass transition temperature of the aerosols.

Besides the numerical results obtained from the model shown in Figure 1-5, the manuscript provides two more useful results: (1) when there is no diffusion limitation within the particle, the gases that have higher saturation mass concentrations will reach gas-particle equilibrium faster; (2) when there is strong diffusion limitation within the particle, gases that have lower volatility may reach gas-particle equilibrium (locally) faster than VOCs with higher volatility. One of the implications is that at a lower temperature (upper troposphere) or when dealing with highly viscous particles, the particle growth process may need to be treated kinetically.

The authors used a numerical model to obtain result #1 above, and it agrees with the findings in Liu et al. 2012 in which an analytical model was used. It is nice to see two different approaches come with the same results and can validate each other. On the other hand, result #2 is more interesting because it shows that local equilibrium could be reached relatively faster for low volatility species when the particle phase is highly viscous. The manuscript also illustrates some concepts that can be commonly misused by folks, such as the difference between equilibrium timescale and the mixing timescale. Because some of the results have been previously discussed in or can be easily inferred from other publications (Liu et al. 2012, Shiraiwa et al. 2011&2012), the novelty of the manuscript needs to be improved. I suggest that the author focus on result #2, which is novel, and use it to make further predictions regarding the physical and chemical processes of aerosols.

Response: We thank Anonymous Referee #1 for the review and the positive evaluation of our manuscript. Following your constructive suggestions, in the revised manuscript, we add (1) a contour plot of $\tau_{eq}$ as a function of bulk diffusivity and volatility to illustrate under what conditions the fast local equilibrium may be achieved to highlight the result #2 you are interested; and (2) simulations for open systems and the results are compared with $\tau_{eq}$ in a closed system. We find partitioning of LVOC is very different in open and closed systems and the corresponding implications in SOA evolution in ambient air and chemical transport models are further broadened. As Referee #2 pointed, this publication essentially closes the loop between predicting phase state and calculating the gas/particle equilibration time. We believe after addition of above two aspects, the novelty of the revised manuscript is improved. Please see the detailed response below.

**Referee Major Comment:**

(1) For instance, what is the relationship between particle sizes and condensation/evaporation kinetics of gases with different volatility interacting with particles with various viscosity?

Response: Thanks for this helpful comment. In our ACPD manuscript, Figure 5 (Figure 6b and d in the revised manuscript) has shown the relationship between $\tau_{eq}$ and particle size for LVOC condensing on less viscous as well as highly viscous particles. In the revised manuscript, we add comparable calculation for SVOC (Fig. 6a, c). This issue has also been discussed in previous studies, e.g., Liu et al. (2013) and Mai et al. (2015). For example, Mai et al. (2015) presented $\tau_{eq}$ as a function of particle diameter and volatility, showing that $\tau_{eq}$ increases as the particle diameter increases or the volatility of the condensing species decreases when particles are liquid with partitioning limited by interfacial transport. When particles are highly viscous with bulk diffusion-limited partitioning, the time to reach full equilibrium depends on mixing timescale. The following discussions have been added in the revised manuscript:

Lines 259-263: "Previous studies have shown that $\tau_{eq}$ depends on particle size (Liu et al., 2013; Zaveri et al., 2014; Mai et al., 2015) and particle mass loadings (Shiraiwa and Seinfeld, 2012; Saleh et al., 2013). For further examination of these effects at different $T$, Figure 6 shows the dependence of $\tau_{eq}$ of SVOC ($C^0 = 10$ μg m$^{-3}$) and LVOC ($C^0 = 0.1$ μg m$^{-3}$) on the mass concentration and the diameter of pre-existing particles".

Lines 271-274: "When particles are less viscous at 298 K ($D_b = 10^{-11}$ cm$^2$ s$^{-1}$) $\tau_{eq}$ of SVOC is shorter than that of LVOC for the same particle size and mass loadings. When partitioning into highly viscous particles at 250 K ($D_b = 10^{-18}$ cm$^2$ s$^{-1}$), SVOC takes longer time than LVOC to reach equilibrium".

Lines 275-285: "Typical ambient organic mass concentrations in Beijing, Centreville in southeastern US, Amazon Basin, and Hyytiälä, Finland are indicated in Fig. 6. The particle phase state was observed to be mostly liquid in highly polluted episodes in Beijing (Liu et al., 2017), under typical atmospheric conditions in the southeastern US (Pajunoja et al., 2016), and under background conditions in Amazonia (Bateman et al., 2017). At these conditions $\tau_{eq}$ should be mostly less than 30 minutes (Fig. 6a, b). Particles were measured to be semi-solid or amorphous solid in clear days in Beijing (Liu et al., 2017), in Amazonia when influenced by anthropogenic emissions (Bateman et al., 2017), and the boreal forest in Finland (Virtanen et a., 2010). Under these conditions and also when particles are transported to the free troposphere, $\tau_{eq}$ can be longer than 1 hour especially in remote areas with low mass loadings (Fig. 6c, d)".

[Figure]

**Figure 6.** Equilibration timescale ($\tau_{eq}$) for (a, c) SVOC ($C^0$ = 10 μg m$^{-3}$) and (b, d) LVOC ($C^0$ = 0.1 μg m$^{-3}$) as a function of particle diameter (nm) and mass concentration (μg m$^{-3}$) of pre-existing particles at 60% RH and $T$ of (a-b) 298 K and (c-d) 250 K in the closed system. The glass transition temperature of pre-existing particles under dry conditions ($T_{g,org}$) is set to be 270 K, which leads to $D_b$ of (a-b) 10$^{-11}$ cm$^2$ s$^{-1}$ and (c-d) 10$^{-18}$ cm$^2$ s$^{-1}$. Ambient organic mass concentrations are indicated with arrows.

(2) I would also be curious to know what is the boundary criteria for result #2 to happen, i.e., how viscous would the particles have to be or how low the volatility of the VOC have to be in order to reach relatively fast local equilibrium? The author can also explore the effects of equilibrium partitioning when the gases can both partition in and react with the particle phase.

Response: This is a very interesting point. To address your question, we conducted additional simulations for $\tau_{eq}$ as a function of bulk diffusivity and volatility for both open and closed systems. The results of such simulations are shown in new Figure 4. The effect of particle-phase reactions on SOA partitioning is an important question, which is beyond the scope of this study. We plan to follow up on this issue in our future study. The following discussions have been added in the revised manuscript:

Lines 194-201: "We further computed $\tau_{eq}$ as a function of $D_b$ and $C^0$ in the closed system. As shown in Fig. 4a, when $D_b$ is higher than ~10$^{-13}$ cm$^2$ s$^{-1}$, $\tau_{eq}$ is insensitive to bulk diffusivity but sensitive to volatility: decreasing volatility increases $\tau_{eq}$ in this regime. In the regime with $D_b$ lower than ~10$^{-13}$ cm$^2$ s$^{-1}$ and $C^0$ higher than ~10 μg m$^{-3}$, $\tau_{eq}$ is controlled by bulk diffusivity: $\tau_{eq}$ increases from 30 s to longer than 1 year as $D_b$ decreases from 10$^{-13}$ cm$^2$ s$^{-1}$ to 10$^{-20}$ cm$^2$ s$^{-1}$. In the regime with $D_b$ < ~10$^{-13}$ cm$^2$ s$^{-1}$

and $C^0 < \sim 10$ µg m$^{-3}$, $\tau_{eq}$ depends on both diffusivity and volatility. Decreasing volatility would lead to shorter $\tau_{eq}$ due to an establishment of local equilibrium of LVOC".

Lines 221-225: "Figure 4b shows simulated evaporation timescales as a function of $D_b$ and $C^0$ in an open system, which agrees very well with Fig. 3 in Liu et al. (2016). It shows that for less viscous particles $\tau_{eq}$ is limited by volatility, while for highly viscous particles $\tau_{eq}$ is insensitive to volatility and controlled by bulk diffusivity".

Lines 127-128: "Particle-phase reactions and their potential impacts on particle visocsty are also not considered in this study".

Lines 361-369: "Incorporation of the particle-phase formation of oligomers and other multifunctional high molar mass compounds can lead to a reduced bulk diffusivity (Pfrang et al., 2011; Hosny et al., 2016), which may prolong the equilibration timescales. Decomposition of highly oxidized molecules (e.g., organic hydroperoxides) in water may also affect gas-particle partitioning (Tong et al., 2016). Current simulations are focused on trace amount of SVOC or LVOC condensing on mono-dispersed particles with negligible particle growth. Potential phase transition in the course of particle growth/evaporation should also be incorporated in future simulations".

[Figure]

**Figure 4.** Contour plot of equilibration timescale ($\tau_{eq}$) as a function of bulk diffusivity ($D_b$) and saturation mass concentration ($C^0$) for (a) condensation in the closed system and (b) evaporation in the open system. The initial mass concentration of pre-existing particles is set to be 20 µg m$^{-3}$ with the number concentrations of $3 \times 10^4$ cm$^{-3}$ and the initial particle diameter of 100 nm. Viscosity is calculated from the Stokes-Einstein equation assuming the effective molecular radius of $10^{-8}$ cm at $T$ of 298 K.

(3) My other question is that most of the modeling results shown the manuscript assumed that the gas-particle is in a closed system. How realistic is the closed system in ambient environment? Would the ambient environment often be an open system for

evaporation kinetics?

Response: Thanks for this helpful comment. To address this question, we add simulations for an open system (Fig. 4b, S5, and S7) in the revised manuscript. The following discussions have been added in the revised manuscript.

Lines 202-208: "In an open system with fixed vapor concentration (Fig. S5), $\tau_{eq}$ of SVOC is slightly longer but on the same order of magnitude as $\tau_{eq}$ in the closed system, as relatively small amounts of SVOC need to condense to reach equilibrium. In contrast, $\tau_{eq}$ of LVOC in the open system become dramatically longer as LVOC continue to condense into the particle phase because of low volatility (Pankow, 1994). For further simulations we focus mainly on the closed system and the corresponding simulations for the open system are provided in the supplement".

Lines 221-225: "Figure 4b shows simulated evaporation timescales as a function of $D_b$ and $C^0$ in an open system, which agrees very well with Fig. 3 in Liu et al. (2016). It shows that for less viscous particles $\tau_{eq}$ is limited by volatility, while for highly viscous particles $\tau_{eq}$ is insensitive to volatility and controlled by bulk diffusivity".

Lines 242-244: "The corresponding simulations of SVOC partitioning in the open system (Fig. S7) show a similar pattern as $\tau_{eq}$ in the closed system".

Lines 306-312: "The timescale of gas-particle partitioning can be different in closed or open systems especially for LVOC (Fig. 4, S7). The closed system simulations represent SOA partitioning in chamber experiments and in closed atmospheric air mass, which could be justified well within seconds-to-minutes timescales and possibly up to hours depending on meteorological conditions. The real atmosphere may be approximated better as an open system due to dilution and chemical production and loss especially at longer timescales".

[Figure]

**Figure S5.** Temporal evolution of mass concentrations of the condensing compound Z in the gas phase ($C_g$), just above the particle surface ($C_s$), and in the particle phase ($C_p$) in the open system. $\tau_{eq}$ is marked with the red circle. RH = 60% and $T$ is (a–b) 298 K and (c–d) 250 K. The $C^0$ of Z is (a, c) 10 µg m$^{-3}$ and (b, d) 0.1 µg m$^{-3}$. The

glass transition temperature of pre-existing particles under dry conditions ($T_{g,org}$) is set to be 270 K, which leads to $D_b$ of (a–b) $10^{-11}$ cm$^2$ s$^{-1}$ and (c–d) $10^{-18}$ cm$^2$ s$^{-1}$. The initial mass concentration of pre-existing particles is set to be 20 μg m$^{-3}$ with the number concentrations of $3 \times 10^4$ cm$^{-3}$ and the initial particle diameter of 100 nm.

[Figure]

**Figure S7.** Equilibration timescale ($\tau_{eq}$) as a function of temperature and relative humidity in the open system. The glass transition temperatures of pre-existing particles at dry conditions ($T_{g,org}$) are (a) 240 K, (b) 270 K, and (c) 300 K, respectively. The saturation mass concentration ($C^0$) of the condensing compound is 10 μg m$^{-3}$ (SVOC). The mass concentration of pre-existing particles is set to be 20 μg m$^{-3}$ with the number concentrations of $3 \times 10^4$ cm$^{-3}$ and the initial particle diameter of 100 nm.

**Referee Minor Comments:**
The author should also include Price et al. 2015 in the reference list in line 115.
Reference: Shiraiwa et al. 2011: doi.org/10.1073/pnas.1103045108
Liu et al. 2012: doi.org/10.1080/02786826.2012.730163
Shiraiwa et al. 2012: doi.org/10.1029/2012GL054008
Price et al. 2015: doi.org/10.1039/C5SC00685F
Response: Price et al. (2015) has been included on Line 123 in the revised manuscript. Besides this, Liu et al. (2013) has been included on Lines 181, 259 and 357.

Liu, C., Shi, S., Weschler, C., Zhao, B. and Zhang, Y.: Analysis of the dynamic interaction between SVOCs and airborne particles, Aerosol Sci. Technol., 47, 125-136, https://doi.org/10.1080/02786826.2012.730163, 2013.

---

## Author Comment (AC2) · 5 Apr 2019

**Response to the comments of Anonymous Referee #2**
**Referee General Comment:**

In this manuscript, the authors use simulations to calculate the gas/particle equilibration time for secondary organic aerosol as a function of temperature, relative humidity, and SOA microphysical properties. The topic of SOA partitioning, equilibration, and phase state are highly relevant in atmospheric chemistry, the topic is timely, and will be of interest to the readers of ACP. This work is a logical extension of the previous work done by the PI. It essentially combines work in predicting the phase state of particles as a function of atmospheric conditions with work on calculating gas/particle equilibration times (Shiraiwa et al., 2017; Shiraiwa and Seinfeld, 2012). Based on past results from the PI on these topics, the present results are not particularly surprising, but I think there is enough new material here to warrant publication. This publication essentially closes the loop between predicting phase state and calculating the gas/particle equilibration time. With that said, there are a few areas that could be improved before publication. The authors could do a better job of calling out, in the manuscript, what is new unique about this manuscript relative to the previous publications by the PI. A few of the conclusions reached in this work also seem to contradict the PI's previous publications and the authors should clear this up. The authors could help readers put this work into context if they explain why they make certain assumptions in their model (i.e., a closed system), offer some insights into how realistic these assumptions are relative to the ambient atmosphere, and explain how their conclusions would be different if/when the assumptions are not atmospherically representative. While the manuscript is generally well written and clear, there was one section that was somewhat confusing and should be clarified before publication. With that said, there are no major shortcomings with the manuscript and, providing the authors make some revisions, I have no reservations about recommending this manuscript for publication in ACP.

Responses: We thank Anonymous Referee #2 for the review and the positive evaluation of our manuscript. As you pointed out, this is the first study to directly relate equilibration timescale of SOA partitioning to ambient temperature and relative humidity, which has important implications in treatment of SOA evolution in chemical transport models. The novelty of the revised manuscript is further strengthened by two additional new results. Firstly, we add a contour plot of $\tau_{eq}$ as a function of bulk diffusivity and volatility to define the regimes of diffusivity-limited and volatility-limited partitioning. Secondly, we add simulations for open systems and the results are compared with $\tau_{eq}$ in closed systems. The implications of $\tau_{eq}$ in closed and open systems are further broadened for SOA evolution in ambient atmosphere and chemical transport models. Following your suggestion, we clarify that apparently contradicting conclusions regarding $\tau_{eq}$ of LVOC actually are consistent with PI's previous publication (e.g., Shiraiwa & Seinfeld, 2012). We also revise the last figure and associated section for better presentation of our results. Please see the detailed response below.

**Referee Major Comment:**

(1) Lines 23-25, Figure A1, and elsewhere. In the present manuscript the authors conclude that the equilibration timescale for low volatile compounds is shorter than for semi-volatile compounds when other conditions are equal. However, Shiraiwa and Seinfeld 2012 report the opposite (see for example Figure 2), with ELVOC's having longer equilibration times than SVOCs (Shiraiwa and Seinfeld, 2012). The authors should comment in the text on why these studies reach opposite conclusions.

Response: The results in this study are actually consistent with Shiraiwa and Seinfeld (2012) even though our previous statements on Lines 23-25 were somewhat misleading. Figure 2 in Shiraiwa and Seinfeld (2012) was presented for liquid particles showing that $\tau_{eq}$ of LVOC is longer, which agreed with the simulations in our current study showing that for less viscous particles LVOC takes longer time than SVOC to reach equilibrium (Fig. 2a-b). Shiraiwa and Seinfeld (2012) did not compare $\tau_{eq}$ of LVOC and SVOC condensing on highly viscous particles, which has been simulated in current study showing that $\tau_{eq}$ of LVOC is shorter (Fig. 2c-d). We clarified this point on Line 23 in the revised manuscript. In addition, we add Fig. 4 in the revised manuscript to systematically evaluate the dependence of $\tau_{eq}$ on both volatility and bulk diffusivity. Please also refer to our response to Comment (2) of Referee #1.

(2) Lines 115-123. I understand that the authors need to make some assumptions or approximations in their model/calculations. I am trying to understand how atmospherically realistic these assumptions are. The two main assumptions in the present model are of a closed system and that the condensation of molecule Z does not alter the composition and microphysical properties of the pre-existing particles. It is clear that the real atmosphere isn't a closed system. The argument could be made that on seconds-to-minutes timescales, it may approximate a closed system, but the processes that authors are modeling are sometimes occurring on timescales of hours or even days. In addition, one of the author's major conclusions is that low volatility material reaches equilibrium more slowly than higher volatility material. I understand why this is the case in a closed system, but would this conclusion also hold in an open system like in the atmosphere with a constant dilution and/or loss of gas-phase molecules? A plume transported from the surface to the upper troposphere would experience an evaporative driving force where this model seems focused on cases where the driving force is toward the particle phase (condensation). With respect to a single compound (Z) changing the composition and microphysical properties, it may be true that a single molecule or even a few molecules rarely make up the bulk of the SOA mass. However, in the real atmosphere, particles obviously grow and their composition and microphysical properties change as SOA condenses. The PI of this manuscript has previously used kinetic modeling to reproduce particle growth, so I'm not really clear on why these assumptions needed to be made (Shiraiwa et al., 2013). In the both cases, I think it is important that the authors explain: why they chose to make these assumptions, how likely it is that these assumptions are representative of the atmosphere, and how their conclusions would likely be different if the assumptions are not correct.

Response: Thanks for this helpful comment. Based on your suggestions we add

simulations for an open system (Fig. 4b, S5 and S7) in the revised manuscript. Figure S5 and S7 show that for condensation of SVOC, $\tau_{eq}$ is slightly longer but on the same order of magnitude as $\tau_{eq}$ in the closed system. In contrast, $\tau_{eq}$ of LVOC condensation in the open system become dramatically longer as LVOC keep condensing into the particle phase because of low volatility. For evaporation in an open system with continuous removal/dilution of gas-phase LVOC, $\tau_{eq}$ of LVOC is also much longer than that in a closed system due to continuous evaporation (Fig. 4b). For the details please refer to the response to Comment (3) of Referee #1. The implications of $\tau_{eq}$ in closed versus open systems in SOA evolution are broadened. In the revised manuscript we state that:

Lines 306-312: "The timescale of gas-particle partitioning can be different in closed or open systems especially for LVOC (Fig. 4, S7). The closed system simulations represent SOA partitioning in chamber experiments and in closed atmospheric air mass, which could be justified well within seconds-to-minutes timescales and possibly up to hours depending on meteorological conditions. The real atmosphere may be approximated better as an open system due to dilution and chemical production and loss especially at longer timescales".

We agree that condensation of substantial amounts of materials may lead to changes in particle microphysical properties including phase state and viscosity, which is beyond the scope of current study, even though this is indeed an important aspect. In this study we let only trace amounts to condense so that physical properties including size and phase state would remain unaffected. As KM-GAP can indeed treat evolution of particle properties upon particle growth/evaporation, we plan to explore this aspect systematically by varying particle-phase reaction rates and resulting impacts on phase state in future studies. Following your suggestion, in the revised manuscript we broaden the discussion as below:

Lines 361-369: "Incorporation of the particle-phase formation of oligomers and other multifunctional high molar mass compounds can lead to a reduced bulk diffusivity (Pfrang et al., 2011; Hosny et al., 2016), which may prolong the equilibration timescales. Decomposition of highly oxidized molecules (e.g., organic hydroperoxides) in water may also affect gas-particle partitioning (Tong et al., 2016). Current simulations are focused on trace amount of SVOC or LVOC condensing on mono-dispersed particles with negligible particle growth. Potential phase transition in the course of particle growth/evaporation should also be incorporated in future simulations".

(3) Lines 276-292 and Figure 5. This figure and associated text was confusing. The figure was confusing because particle diameter and total mass loading are typically not independent of one another in a model or in the atmosphere. I eventually understood the point the authors were trying to make. Perhaps providing some context by pointing out where different atmospheric regimes lie (i.e., remote, typical continental, polluted) in the figure would help. The associated text is also confusing; it wasn't clear what point the authors were trying to make here. They seem to postulate several different processes which could determine particle equilibration

timescales (e.g., bulk diffusion, gas-diffusion). Can't the KM-GAP model be used to clear this up? Overall, I'm not sure what message the authors are trying to convey here.

Response: Following your suggestion, we indicate typical ambient organic mass concentrations in Beijing (Liu et al., 2017), southeastern US (Pajunoja et al., 2016), Amazon Basin (Bateman et al., 2017), and Hyytiälä, Finland (Virtanen et a., 2010) in Fig. 6, where ambient phase state measurements are available. Figure 6 indeed has implications on how different ambient conditions have effect on SOA partitioning. For clarification, the following discussions have been added in the revised manuscript.

Lines 259-261: "Previous studies have shown that $\tau_{eq}$ depends on particle size (Liu et al., 2013; Zaveri et al., 2014; Mai et al., 2015) and particle mass loadings (Shiraiwa and Seinfeld, 2012; Saleh et al., 2013). For further examination of these effects at different $T$…".

Lines 275-285: "Typical ambient organic mass concentrations in Beijing, Centreville in southeastern US, Amazon Basin, and Hyytiälä, Finland are indicated in Fig. 6. The particle phase state was observed to be mostly liquid in highly polluted episodes in Beijing (Liu et al., 2017), under typical atmospheric conditions in the southeastern US (Pajunoja et al., 2016), and under background conditions in Amazonia (Bateman et al., 2017). At these conditions $\tau_{eq}$ should be mostly less than 30 minutes (Fig. 6a, b). Particles were measured to be semi-solid or amorphous solid in clear days in Beijing (Liu et al., 2017), in Amazonia when influenced by anthropogenic emissions (Bateman et al., 2017), and the boreal forest in Finland (Virtanen et a., 2010). Under these conditions and also when particles are transported to the free troposphere, $\tau_{eq}$ can be longer than 1 hour especially in remote areas with low mass loadings (Fig. 6c, d)".

[Figure]

**Figure 6.** Equilibration timescale ($\tau_{eq}$) for (a, c) SVOC ($C^0 = 10$ μg m$^{-3}$) and (b, d) LVOC ($C^0 = 0.1$ μg m$^{-3}$) as a function of particle diameter (nm) and mass concentration (μg m$^{-3}$) of pre-existing particles at 60% RH and $T$ of (a-b) 298 K and (c-d) 250 K in the closed system. The glass transition temperature of pre-existing particles under dry conditions ($T_{g,org}$) is set to be 270 K, which leads to $D_b$ of (a-b) 10$^{-11}$ cm$^2$ s$^{-1}$ and (c-d) 10$^{-18}$ cm$^2$ s$^{-1}$. Ambient organic mass concentrations are indicated with arrows.

**Referee Minor Comments and Technical Corrections:**

(1) Line 89. It is a little confusing here about what temperature was used in the calculations. I wasn't clear whether 273.15 K was used (the most common definition of standard temperature), or if the temperature was variable as a function of pressure altitude. Adding to the confusion, Table S2 lists standard temperature as 288.15 K. Please clarify.

Response: In our simulations the temperature is varied from 220 K to 310 K (Fig. 5) while atmospheric pressure is calculated as a function of $T$ based on the International Standard Atmosphere (ISA): $P/P_{standard} = (T/T_{standard})^{g/LR}$, where $P_{standard}$ and $T_{standard}$ are standard sea level $P$ and $T$ in ISA, and L is the lapse rate of 6.5 K/km in the troposphere. This has been clarified on Lines 94-95 and Table S2 in the revised manuscript.

(2) Lines 108-110. An assumption here is that the organic and aqueous phases are not phase separated. The authors point out later in the manuscript that phase separations may occur, but I suggest also briefly mentioning that phase separation has been observed for laboratory generated SOA (You et al., 2012) here, since it is very relevant to their modeling results.

Response: The following sentence has been added on Lines 116-119 in the revised manuscript:

"For simplicity we assume particles are ideally-mixed, even though phase-separated particles are observed for ambient and laboratory generated SOA particles under certain conditions (You et al., 2012; Renbaum-Wolff et al., 2016)".

(3) Lines 144 and elsewhere. I found the use of $C_0$ as an abbreviation more confusing than necessary. It seems $C_0$ is identical to the much more commonly used $C^*$. Why not use the commonly accepted $C^*$? The authors also use $C_{p,0}$ and $C_{g,0}$, which have a different meaning and cause some confusion with $C_0$. Whatever symbol the author use for the saturation vapor pressure please define it the first time it is used.

Response: Instead of $C_0$, $C^0$, which is commonly used for the pure compound saturation mass concentration, is used throughout the revised manuscript. The effective saturation mass concentration $C^*$ is not used as it includes the effect of non-ideal thermodynamic mixing which is not considered in this study. Lines 154-157 have been re-written in the revised manuscript as:

"Figure 2a presents simulations for a semi-volatile organic compound (SVOC) with the pure compound saturation mass concentration ($C^0$) of 10 μg m$^{-3}$ condensing on particles with $D_b$ of $10^{-11}$ cm$^2$ s$^{-1}$ at RH = 60% and $T$ = 298 K (Fig. S2)".

(4) Figures 4, A1, A2. The labels on different contours were illegible on the printed document.

Response: The resolution of the figures has been improved in the revised manuscript.